# Rapid rebalancing of co-tuned ensemble activity in the auditory cortex

HiJee Kang[1], Travis A Babola[1], Patrick O Kanold[1,2]*

[1]Department of Biomedical Engineering, Johns Hopkins University, Baltimore, United States; [2]Kavli Neuroscience Discovery Institute, Johns Hopkins University, Baltimore, United States

## eLife Assessment

This study uses all-optical electrophysiology methods to provide a **valuable** insight into the organization of cortical networks and their ability to balance the activity of groups of neurons with similar functional tuning. The all-optical approach used in this study is impressive, and the claim that the effects of optical stimulation correspond to a specific homeostatic mechanism is **solid**. The work will be of interest to neurobiologists and to developers of optical approaches for interrogating brain function.

*For correspondence:
pkanold@gmail.com

**Competing interest:** The authors declare that no competing interests exist.

**Abstract** Sensory information is represented by small varying neuronal ensembles in sensory cortices. In the auditory cortex (AC), repeated presentations of the same sound activate differing ensembles, indicating high trial-by-trial variability in activity even though the sounds activate the same percept. Efficient processing of complex acoustic signals requires that these sparsely distributed neuronal ensembles actively interact in order to provide a constant percept. Thus, the differing ensembles might interact to process the incoming sound inputs. Here, we probe interactions within and across ensembles by combining in vivo two-photon $Ca^{2+}$ imaging and holographic optogenetic stimulation in awake mice to study how increased activity of single cells affects the cortical network. We stimulated a small number of neurons sharing the same frequency preference alongside the presentation of a target pure tone, further increasing their tone-evoked activity. We found that other non-stimulated co-tuned neurons decreased their tone-evoked activity when the frequency of the presented pure tone matched their tuning property, while non-co-tuned neurons were unaffected. Activity decrease was greater for non-stimulated co-tuned neurons with higher frequency selectivity. Co-tuned and non-co-tuned neurons were spatially intermingled. Our results show that co-tuned ensembles communicated and balanced their total activity across the larger network. The rebalanced network activity due to external stimulation remained constant. These effects suggest that co-tuned ensembles in AC interact and rapidly rebalance their activity to maintain encoding network dynamics, and that the rebalanced network is persistent.

## Introduction

Sensory perception requires fast encoding of relevant stimuli from a mixture of complex signals. Sensory cortices play a vital role in such sensory processing. In the auditory domain, for example, small neuronal ensembles in the auditory cortex (AC) are actively engaged to efficiently perceive relevant acoustic information (*Read et al., 2001*; *Schnupp et al., 2001*; *Lu et al., 2018*; *Fritz et al., 2003*). The AC contains multiple ensembles of neurons that can be functionally identified, for example, those formed by subsets of neurons preferring the same frequency also referred to as co-tuned neurons (*Bandyopadhyay et al., 2010*; *Rothschild et al., 2010*; *Panzeri et al., 2015*). Repeated presentation

of the same acoustic stimulus such as a tone of the same frequency leads to a stable percept. But in the AC, different ensembles of neurons are activated together at each repeat indicating a high trial-by-trial variability (*Bandyopadhyay et al., 2010*; *Bowen et al., 2020*; *Francis et al., 2018*; *Francis et al., 2022*). Activation of these different subsets of co-tuned neurons at each presentation of a stimulus reflects a sparse encoding of sound stimuli. Such sparse representation of co-activated neurons enables efficient coding with reduced metabolic energy to process complex information (*Hromádka et al., 2008*; *Hromádka and Zador, 2009*; *Liang et al., 2019*; *Yu and Yu, 2017*; *Jadi and Sejnowski, 2014*; *Grabner et al., 2006*). The sparse neuronal representation raises key questions of how activation of different ensembles leads to the same percept and how the overall activity within the cortical network is balanced across ensembles of co-tuned neurons. In particular, when a specific sound is present, a subset of co-tuned neurons will be activated, but not all co-tuned neurons (*Bowen et al., 2019*). Given that the percept of a repeating stimulus is constant, we speculated that neural activity is balanced across co-tuned as well as non-co-tuned ensembles.

While neuronal ensembles constantly update their activities based on incoming information, how the activation of a particular sparse neuronal ensemble affects other neurons within the network to maintain the overall network balance for processing specific sensory information in vivo is largely unknown. In vivo optogenetic stimulation studies in the visual cortex (VC) suggested that inhibitory processes play a role in balancing network activity. In particular, in vivo single-cell holographic stimulation on a group of target cells, which induced increased response amplitude, resulted in changes in the response amplitudes of neighboring non-target neurons in the primary VC (V1) (*Chettih and Harvey, 2019*), with similarly tuned neurons' activity being suppressed. Moreover, in vivo holographic optogenetic stimulation showed that visually suppressed neurons had attenuated response amplitudes when holographic stimulation was given along with the visual stimulus presentation, which was not observed in visually activated neurons (*LaFosse et al., 2024*). This suggested that neurons exhibit supralinear-to-linear input–output (IO) functions in vivo, rather than threshold-linear IO functions observed in vitro. These studies indicate that inhibitory influence from additional neuronal activation in the VC seems to play a major role during in vivo sensory processing, likely to maintain the activity balance of the network by modulating activities of neighboring neurons that share a similar tuning property.

One major difference between VC and AC is that the frequency tuning of neurons in the AC is less spatially localized, especially in a superficial layer (layer 2/3) (*Winkowski and Kanold, 2013*). The local frequency preferences in the AC are diverse, thus neighboring neurons can show widely differing tuning properties (*Bandyopadhyay et al., 2010*). To test how activity in specific AC cells among an intermingled and spatially distributed co-tuned and non-co-tuned cell population is balanced during auditory processing, we stimulated a small group of AC cells using in vivo holographic optogenetic stimulation (*Adesnik and Abdeladim, 2021*; *Yang and Yuste, 2021*) while imaging AC population activity using two-photon $Ca^{2+}$ imaging in awake mice. We further tested whether any activity changes induced by holographic stimulation persist, as recurrent cortical networks engage homeostatic plasticity to stabilize overall network activity levels (*Karmarkar and Buonomano, 2007*; *Liu et al., 2023*). Stimulating small ensembles of co-tuned neurons together with the presentation of a pure tone in their preferred frequency increased their tone-evoked activity. Furthermore, we observed that non-stimulated co-tuned neurons decreased their tone-evoked activity. Non-co-tuned ensembles did not exhibit such changes in tone-evoked responses, regardless of the pure tone frequency. Thus, the increased activity in the stimulation-targeted ensemble had caused a decrease in activity in the non-stimulated co-tuned ensembles, specifically when the stimulation-paired pure tone was their preferred frequency. Non-target co-tuned neurons exhibiting such effects were not necessarily neighboring the targeted cells, suggesting specific interactions between co-tuned but not co-located neurons. Lastly, the decreased activity in the non-stimulated co-tuned ensembles persisted in the subsequent imaging session, even in the absence of holographic stimulation. These results suggest that co-tuned ensembles form interacting overall networks that balance their activity.

## Results

### Optogenetic holographic stimulation increases activity in small ensembles in vivo

To sparsely manipulate neuronal ensembles, we used in vivo holographic stimulation. To achieve reliable and selective in vivo holographic optogenetic stimulation of small ensembles of neurons with single-cell precision, we generated an AAV co-expressing the red-shifted opsin rsChRmine and GCaMP8s (AAV9-hSyn-GCaMP8s-T2A-rsChRmine), as rsChRmine minimizes the optical cross-talk reducing a possible activation from the imaging laser (940 nm excitation wavelength) (*Kishi et al., 2022*). Injecting AAV9-hSyn-GCaMP8s-T2A-rsChRmine into AC yielded cells expressing both GCaMP and opsin (*Figure 1A*). We first tested the efficiency and reliability of holographic stimulation by targeting either a single cell or a small ensemble of five cells. For single-cell stimulation, we varied the stimulation point from the target cell position by 10, 20, and 30 µm along either the x-axis or y-axis of the fields of view (FOV; n=15 cells, three animals). This resulted in a rapid decay of response amplitudes to stimulation by the distance shift from the original cell position, confirming reliable holographic stimulation at the single-cell level (~15 µm diameter) (mixed-effect model, p<0.05; *Figure 1B*). Furthermore, the stimulation effect of target cells was specific to the targeted z-plane, showing no stimulation effect when the stimulation z-plane was off by 20 µm (*Figure 1—figure supplement 1*). For five-cell stimulation, a majority of cells reliably responded to photo-stimulated in vivo (5 mW/cell, 15 µm spiral, 30 revolutions, 6 s inter-stimulus interval [ISI], 5 trials) and exhibited robust $Ca^{2+}$ responses (*Figure 1C and D*; permutation test, all p<0.05), comparable with responses to other opsins (*LaFosse et al., 2023*; *Sridharan et al., 2022*; *Mardinly et al., 2018*; *Daie et al., 2021*). Thus, in vivo holographic stimulation enables precise targeting and activation of groups of single neurons in AC.

### Optogenetic holographic stimulation increases sound-evoked activity in A1 ensembles

Since repeated sound stimulation activates different ensembles while resulting in the same percept, we reasoned that ensembles interacted and speculated that increased activity in one ensemble would prevent or reduce activity in co-tuned ensembles. We thus next sought to investigate how increased neural activity in small co-tuned ensembles during sound presentation affected sound-evoked responses in targeted and non-target co-tuned and non-co-tuned ensembles. To achieve this, we first needed to identify the tuning properties of single neurons and then target a subset of co-tuned neurons for stimulation. To study how the increased activity of a small number of neurons influences the activity of other neurons according to their frequency tuning properties, we designed an experimental paradigm comprising four sequential imaging sessions (*Figure 1E*).

First, in the cell selection session (*Figure 1E*), we identified tuned ensembles in primary AC (A1) layer 2/3 (L2/3) by assessing frequency tuning properties of neurons within the FOVs covering 550 µm² (total cells = 7344, sound-responsive cells = 1331, FOVs = 23; *Figure 1E*). We presented pure tones of three different frequencies spanning the hearing range of the mouse (4, 16, and 54 kHz, 100 ms duration, 2 s ISIs, 10 repeats for each frequency). We chose 16 kHz and 54 kHz as the representative target ensemble tone frequencies, as 16 kHz is within the most sensitive frequency range of mice (*Kane et al., 2012*) and 54 kHz is within the range of mouse ultrasonic vocalization (*Branchi et al., 2001*). By selecting target ensembles in two different frequencies, we ensured that effects of stimulation were not specific to one particular population. For each condition (16 or 54 kHz target ensemble for stimulation), we selected five target cells to stimulate. To ensure that all cells in the ensemble were selective for the target tone, we chose the most responsive cells in each condition. Thus, for the 16 kHz target ensemble condition, we selected five cells (*target cells*) among the top 30% most responsive cells to the 16 kHz tone. Similarly, for the 54 kHz target ensemble condition, we selected five of the top 30% most 54 kHz tone responsive cells. By selecting target cells sharing the same frequency preference, we aimed at investigating how activity changes from co-tuned neuronal ensembles alter the processing of the target frequency in other co-tuned and non-co-tuned cells.

Second, in the baseline session (*Figure 1E*), we determined the sound-evoked responses of all imaged cells by presenting a series of 16 and 54 kHz pure tones in a random order (100ms duration, 5.8–6.5 sc ISI; baseline session, 30 repeats for each frequency). Exemplar responses of cells from a 16 and 54 kHz ensemble are shown as black traces in *Figure 2A and C*.

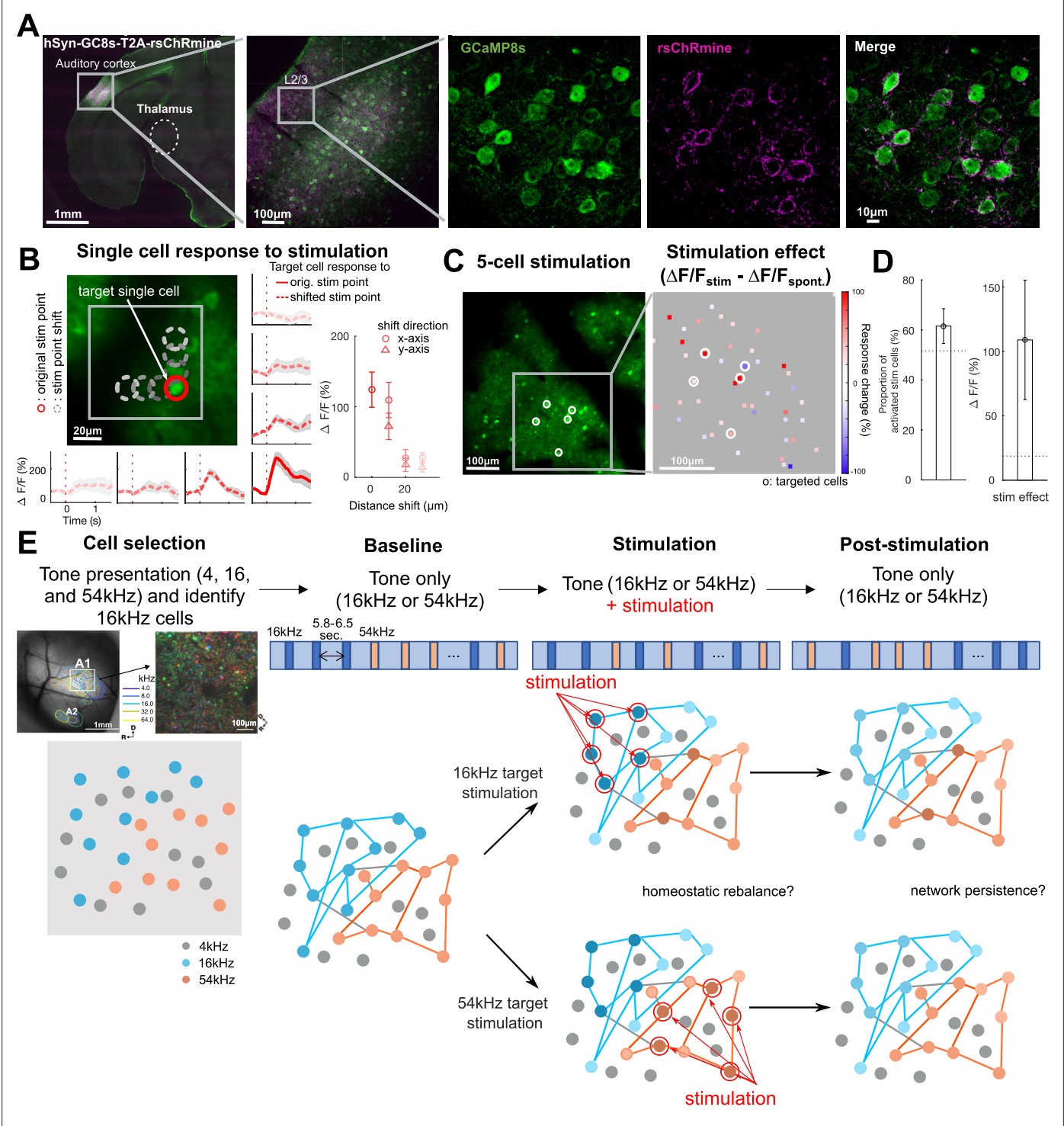

**Figure 1.** Holographic optogenetic stimulation in auditory cortex (AC) and experimental procedure. (**A**) An example brain slice showing cells in AC expressing both GCaMP and opsin (AAV9-hSyn-GC8s-T2A-rsChRmine). (**B**) An example field of view (FOV) where single-cell targeting precision was tested and response traces to the holographic stimulation from an example cell. Stimulation was offset from the original position (red circle) to distance-shifted positions in 10 μm increments (gray dashed circles in the x-axis or y-axis of the FOV). Responses were the greatest when the stimulation was performed on the original cell position (red solid line trace). Rapid amplitude decay along the position shift was observed (red dashed line traces). Gray error shades indicate SEM across trials. A right inset error bar plot shows a grand average amplitude change per stimulation point across all tested cells (n=15 cells, three animals). Error bars indicate SEM across cells. (**C**) An example FOV showing a population of cells (left) and amplitude changes to five-cell stimulation as a stimulation effect ($\Delta F/F_{stim} - \Delta F/F_{spont.}$, right). Filled squares indicate each cell. White circles indicate stimulation-targeted cells. (**D**) (left)

*Figure 1 continued on next page*

*Figure 1 continued*

Proportion of stimulated cells that showed an increase in fluorescence following photostimulation. Error bars indicate SEM across FOVs. A horizontal dashed line indicates average permutation results (random permutation test on 100 iterations, p<0.0001). (Right) Grand average of the stimulation effect across imaging sessions. Error bars indicate SEM across FOVs. A horizontal dashed line indicates average permutation results (random permutation test on 100 iterations, p<0.0001). (**E**) Experimental procedure. A total of four consecutive imaging sessions were acquired: (1) a cell selection session to identify neurons selective for 16 kHz pure tones, (2) a baseline imaging session to acquire tone-evoked activity response to either 16 or 54 kHz pure tone, (3) a stimulation session representing five cells of either 16 kHz or 54 kHz responsive cells as target stimulation to examine the effect of stimulation synchronized to tone presentations, and (4) a post-stimulation session to examine network persistence after stimulation-related changes.

The online version of this article includes the following figure supplement(s) for figure 1:

**Figure supplement 1.** Holographic optogenetic stimulation effectively activates target points.

Third, in the stimulation session (*Figure 1E*), we examined how all sound-responsive cells change their responses when a small group of cells in the network increases their activity. We presented the same tones (16 and 54 kHz in a random order), in tandem with the optogenetic stimulation of five target cells (stimulation session, 100 ms stimulation duration). We performed different sessions for the 16 and 54 kHz target ensembles, varying FOVs for each session (18 FOVs for 16 kHz target ensemble condition and 15 FOVs for 54 kHz target ensemble condition). *Figure 2A and C* show two example FOVs with targeted neurons for a 16 and 54 kHz ensemble, respectively.

Since both imaging and optogenetic stimulation involve optomechanical components, we wanted to ensure that effects were not due to artifacts caused by our stimulation or imaging setup. Moreover, cells can adapt their responses to repeated sound presentation. Thus, to confirm any response changes observed from the stimulation session are due to the optogenetic stimulation rather than simple response change due to acoustic sound presentation, we added an additional control condition. For this control condition, we performed the 'stimulation' session with five target cells but with 0 mW laser power (i.e., no stimulation) to verify that any response changes occurring in the stimulation session compared to the baseline session were not simply due to the eventual response adaptation of neurons to the tuned frequency (control condition; 13 FOVs). By selecting cells and presenting 0 mW laser power, instead of no target cell selection or selecting any other no-cell area within the FOV, we ensured that the laser power given to selected cells was the only difference between the actual stimulation and control conditions, keeping any noise caused by the imaging and stimulation setup the same.

Fourth, after the stimulation session (*Figure 1E*), we performed an additional imaging session (post-stimulation session), presenting another series of 16 and 54 kHz pure tones in a random order to examine whether changes in the sound-evoked responses persisted or reverted back after the stimulation session.

## Optogenetic holographic stimulation increases activity in targeted ensembles

We first identified the effect of the optogenetic stimulation on the targeted ensembles. *Figure 2A and C* show fluorescence traces of exemplar cells from 16 and 54 kHz target ensembles. Optogenetic stimulation increased the sound-evoked fluorescence amplitude in these individual cells. To quantify the effect of the optogenetic stimulation, we compared the tone-evoked fluorescence responses of the targeted cells with and without stimulation (Stimulation effect $= \Delta F/F_{(stimulation\ session)} - \Delta F/F_{(baseline\ session)}$). Around 72% of target cells (66 out of 90 cells over 18 FOVs for 16 kHz target stimulation and 42 out of 60 cells 15 FOVs for 54 kHz target stimulation) showed increased response amplitude during the stimulation session compared to the baseline session, regardless of the tone presented (*Figure 2B and D*; permutation tests, all p<0.001). These results indicate that holographic stimulation was able to reliably increase activity in small populations of neurons. Moreover, given that the target cells we selected were most responsive to their preferred tone frequency, this increase in fluorescence indicates that the cells' responses to their preferred tone were not saturated.

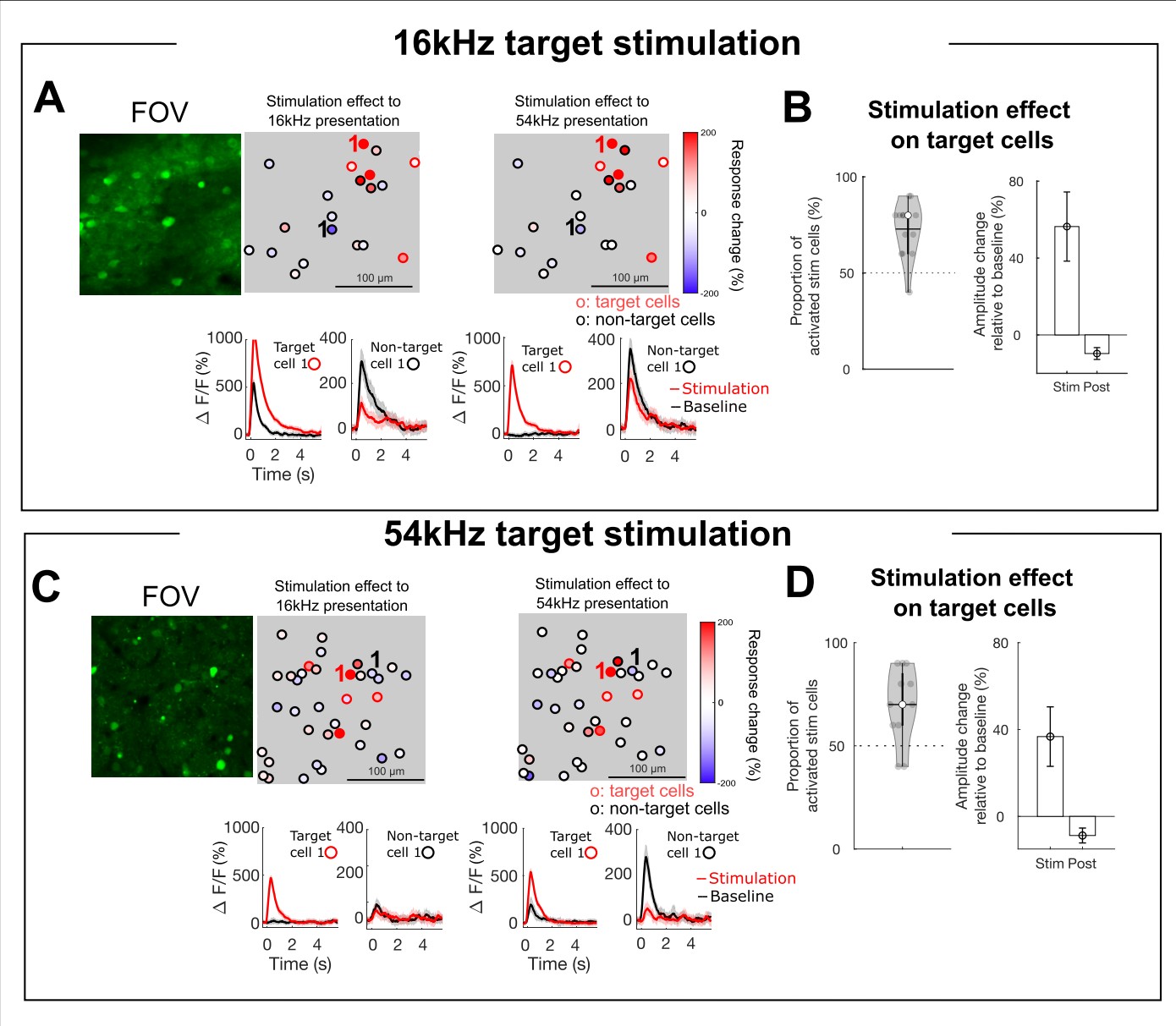

**Figure 2.** Targeted cells and non-target cells show response changes due to stimulation. (**A**) (top) An example field of view (FOV) showing the stimulation effect ($\Delta F/F_{stim} - \Delta F/F_{baseline}$) of sound-responsive cells for 16 kHz target cell stimulation (filled squares). Black circles indicate stimulated target cells. (Bottom) Mean response traces of an example target and non-target cell in baseline session (black) and stimulation session (red). Error bars indicate SEM across trials. An example target cell shows an increased response due to the stimulation. An example non-target cell shows a decreased response due to stimulation on the target cells. (**B**) (left) A violin plot of the proportion of stimulated cells that showed increased activity due to stimulation across FOVs. Horizontal solid line indicates mean proportion, empty circle indicates median proportion, and gray filled circles indicate individual FOVs. (Right) Mean amplitude changes of target cells for stimulation session and post-stimulation session normalized to the baseline session. Error bars indicate SEM across cells. Dashed horizontal lines on both panels indicate average permutation results (random permutation test on 100 iterations, p<0.0001). (**C, D**) Same as ABC for 54 kHz target cell stimulation.

## Optogenetic holographic stimulation decreases activity in non-target co-tuned ensembles

We next investigated whether the optogenetically enhanced sound-evoked activity of a small group of cells would cause activity changes in other non-stimulated cells. During holographic optogenetic stimulation of the targeted cells, the non-target, but sound-responsive cells (n=995 cells for 16 kHz target ensemble condition and n=675 cells for 54 kHz target ensemble condition), also changed

their activity, showing either increased or decreased response amplitudes (*Figure 2A* bottom and C bottom).

If cortical networks rebalance their activity, we speculated that the increased tone-evoked activity in the targeted ensemble would lead to a decrease in tone-evoked activity in coupled ensembles. Such rebalancing would keep the activity within the cortical network stable. Moreover, given that we increased the activity to the preferred sound frequency, if this rebalancing happens, it should occur only for the distinct sound frequency related to the cell's tuning property. For example, stimulation of a 16 kHz ensemble should cause a greater reduction in the 16 kHz tone response of non-targeted 16 kHz cells compared to their response to the 54 kHz tone.

To address these questions, we investigated whether increased activity in the targeted cells influenced the activity of non-target cells and how these changes were related to the tuning properties of the cells. We first confirmed that the overall population activity from sound-responsive cells, including both target and non-target cells, did not differ across conditions (control, 16 kHz, or 54 kHz target ensemble conditions; all $P>0.05$, *Figure 3A*). This suggests that non-target cells may adjust their activities during the target cell stimulation to maintain the global network activity level. To identify the activity changes based on functional characteristics of cells, we defined each sound-responsive cell's frequency selectivity by computing a difference between response amplitude to 16 and 54 kHz from the baseline session $\left(\text{frequency preference} = \left(\Delta F/F_{(16kHz)}\right) - \left(\Delta F/F_{(54kHz)}\right)\right)$. We then divided these cells into either 16 kHz preferring or 54 kHz preferring groups, taking 0 (i.e., no preference) as a criterion (*Figure 3B*). Both subgroups exhibited stronger tone-evoked responses to their preferred frequency, independent of the condition ($t(5700)=4.79$, $p<0.0001$; *Figure 3—figure supplement 1*). This confirms that the criterion for cell group threshold is valid.

We then focused on our main question by comparing the stimulation effect of the two target ensemble groups to the control condition to identify whether stimulation decreased the response of non-target co-tuned neurons. Neural activity in AC rapidly shows stimulus-specific adaptation to the repeated presentation of the stimulus (*Ulanovsky et al., 2004*; *Yarden and Nelken, 2017*; *Malmierca et al., 2014*), which can obscure stimulation-related changes. We thus used the response amplitude change between the baseline and the 'stimulation' control session as a representative threshold to test the effect of the stimulation. We once again used the difference in response amplitude between the baseline and stimulation sessions as the measure of the stimulation effect $\left(\Delta F/F_{(\text{stimulation session})} - \Delta F/F_{(\text{baseline session})}\right)$. Neighboring cells within 20 μm from the target stimulation point were removed from the analysis since they could have been directly affected by the stimulation.

We compared the stimulation effect between non-target co-tuned and non-co-tuned cells across conditions (16 and 54 kHz target ensembles as well as control conditions) for different pure tone presentations. Since our primary interest was how non-target cells respond to increased activity in target ensembles, we focused on conditions where the pure tone frequency matched or did not match the tuning properties of the non-target cells. Since we stimulated during tone presentation the effects of the holographic stimulation and stimulus-specific adaptation co-occurred. To isolate these components, we used a linear mixed-effect model with cell group, condition, and pure tone frequency as fixed factors, and FOVs as a random factor. We then performed analysis of variance (ANOVA) on the model to assess the main effects and interactions.

A marginally significant main effect of the condition ($F(2,37.1)=2.983$, $p=0.0628$) on the response change in the stimulation session relative to the baseline session (i.e., stimulation effect) was observed, indicating that these changes may depend on the stimulation condition. We further analyzed the data to better understand how the different factors interacted in the response amplitude changes. A significant interaction between the pure tone frequency and cell group ($F(1,4397.6)=186.967$, $p<0.0001$) suggests that each cell group responded differently to the two pure tone frequencies. Specifically, the response amplitude decreases in the stimulation session relative to the baseline session were more pronounced for each cell group when the played pure tone matched to their tuning property. This interaction between pure tone frequency and cell group highlights the importance of frequency tuning in modulating response amplitudes. Such response amplitude decreases of non-target cells to their preferred pure tone presentation further align with the stimulus-specific adaptation due to repeatedly presented pure tones (*Ulanovsky et al., 2004*). Additionally, a significant three-way interaction across condition, cell groups, and pure tone frequency ($F(2,4397.6)=3.517$, $p=0.0298$) suggests

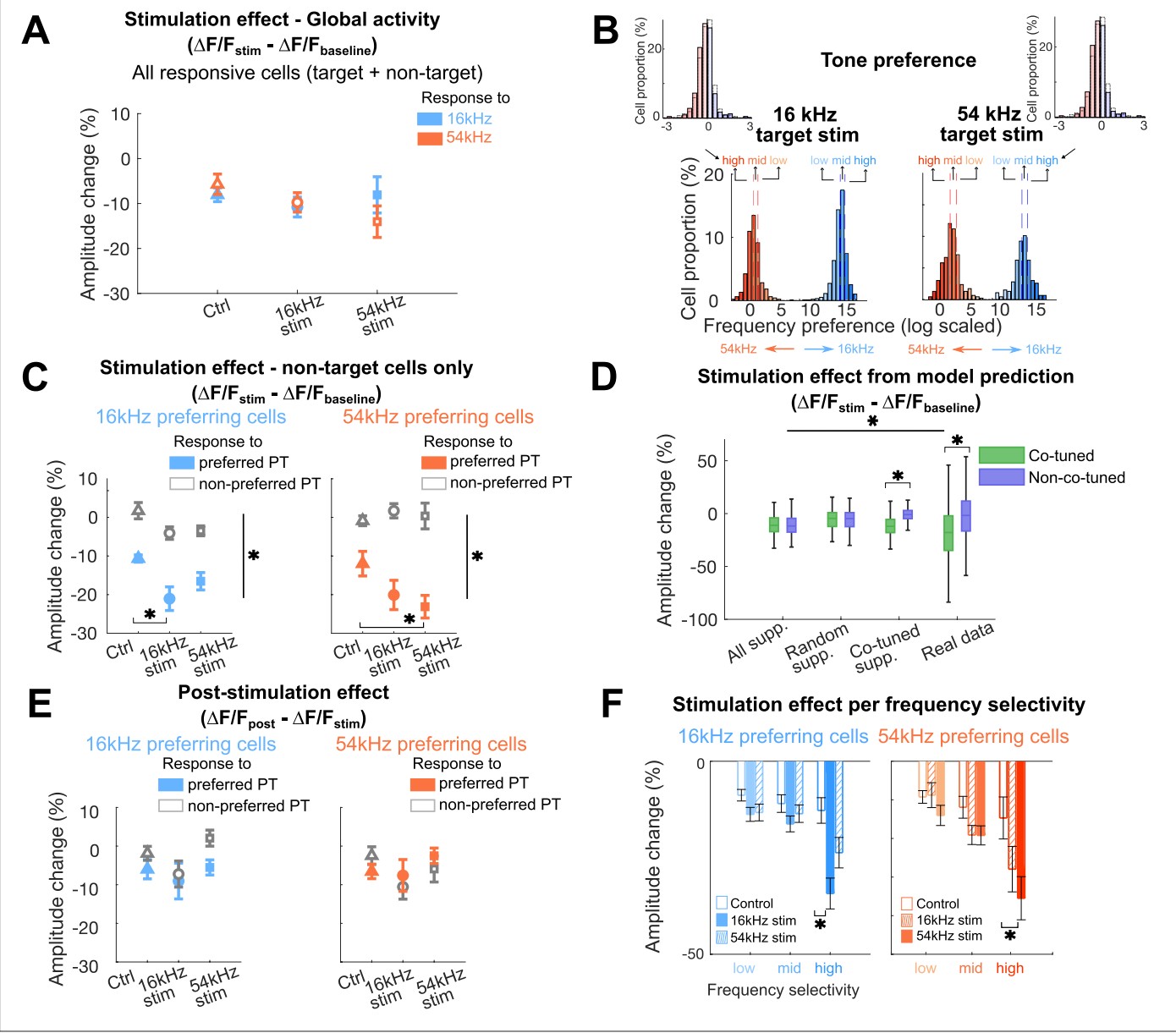

**Figure 3.** Non-target co-tuned cells show more decreased response amplitudes due to stimulation when synchronized with their preferred tones. (**A**) Stimulation effect ($\Delta F/F_{stim} - \Delta F/F_{baseline}$) in all sound-responsive cells, including both target and non-target cells, responding to either 16 kHz (blue) or 54 kHz (orange) pure tones, representing global activity changes due to the stimulation. No significant differences between stimulation conditions and responses to different frequencies were observed (all p>0.05). (**B**) Sub-categorization of cells based on the frequency selectivity for each target stimulation condition (left: 16 kHz stim, right: 54 kHz stim). Cells were first grouped into either 16 kHz preferring cells (blue) or 54 kHz preferring cells (orange). Within each cell group, cells were further subdivided into low-, mid-, and high-frequency selectivity categories based on their 33% quartile ranges. For visualization, frequency preference was log-transformed; original frequency selectivity distributions are shown in the upper insets. Vertical dashed lines indicate 33% quartile ranges. (**C**) Stimulation effect ($\Delta F/F_{stim} - \Delta F/F_{baseline}$) in 16 kHz (blue) and 54 kHz (orange) preferring cells. Both cell groups show decreased amplitude to their preferred frequency regardless of conditions due to acoustic stimulus-specific adaptation. Only co-tuned cells (16 kHz preferring cells for 16 kHz stimulation or 54 kHz preferring cells for 54 kHz stimulation) show a further decrease in response amplitudes due to the stimulation, when the preferred pure tone (PT) frequency was synchronized. Error bars indicate SEM across FOVs (*p<0.0001). (**D**) Stimulation effect from the model prediction. Amplitude changes computed from simulated data by applying cell suppression to all cells (All supp.), random cells (Random supp.), or only co-tuned cells (Co-tuned supp.) were compared with real data. Only the Co-tuned supp. model showed a significant amplitude decrease for co-tuned neurons compared to non-co-tuned neurons, similar to the result from the real data (p<0.05; see text for more detail). (**E**) Post-stimulation effect ($\Delta F/F_{post} - \Delta F/F_{stim}$) 16 kHz (blue) and 54 kHz (orange) preferring cells. No significant response amplitude changes were observed. Error bars indicate SEM across FOVs. (**F**) Response amplitude change based on the frequency selectivity category for each cell group (blue: 16 kHz preferring

*Figure 3 continued on next page*

*Figure 3 continued*

cells, orange: 54 kHz preferring cells). Significant response amplitude changes relative to the control condition were observed only for high-frequency selectivity category when target stimulated cells were co-tuned (*p<0.05).

The online version of this article includes the following figure supplement(s) for figure 3:

**Figure supplement 1.** Subgrouped non-target cells show the frequency preference.

**Figure supplement 2.** Stimulation effect.

the combined effects of the stimulation condition and the cell group on response amplitude depend on the pure tone frequency. The stimulation effect is not uniform across cell groups and depends heavily on the frequency, highlighting a complex interplay between the tuning property of cells, stimulation condition, and presented pure tone frequency.

Consequently, we analyzed post hoc comparisons estimated marginal means with contrasts, as our focus was how co-tuned cells change their responses due to the increased activity in the target cells along with the frequency of the presented pure tone.

For the 16 kHz preferring cell group (n=537), we observed a greater stimulation effect (i.e., decrease in response amplitude) for 16 kHz tone presentation when the 16 kHz target ensemble was stimulated compared to the control condition ($t(124) = 3.114$, p=0.0064). For all other pairs, no significant stimulation effect was observed. This suggests that non-target 16 kHz co-tuned cells reduce their response amplitudes when target ensembles share the same tuning property. Furthermore, such response change occurs only when they process their preferred frequency (*Figure 3C*, left).

We repeated the experiments and the analysis with 54 kHz cells as the target group. In general, we observed similar results. The stimulation effect was significantly more pronounced for 54 kHz tone presentation when 54 kHz target ensemble (n=359) was stimulated compared to the control condition ($t(168) = 3.074$, p=0.0069; all p-values were adjusted for multiple comparisons using the Tukey method). All other pairs did not show any stimulation effect (*Figure 3C*, right).

To further explore whether the stimulation effect could be explained by activity rebalancing within the co-tuned network, we implemented a simple model in which a suppression term was applied either to all non-target cells, randomly selected non-target cells, or specifically to non-target co-tuned cells. By comparing three different model outcomes and the real data, we observed a significant effect of the model type ($F(3, 3343)=56.243$, p<0.0001). Moreover, an interaction between the model type and cell groups was observed ($F(3, 3343)=49.635$, p<0.0001). Applying suppression to only non-target co-tuned cells during the stimulation session yielded a significant response amplitude decrease for co-tuned cells compared to non-co-tuned cells ($F(1, 3343)=48.68$, p<0.0001), which resembles the real data. In contrast, applying suppression to all non-target cells and random non-target cells led to similar amplitude changes in both co-tuned and non-co-tuned neurons ($F(1, 3343)=0.01$, p=0.925 for all suppression; $F(1, 3343)=0.05$, p=1), which was not observed in either the real data or the simulated data restricted to co-tuned cell suppression (*Figure 3D*). These results suggest that the target cell stimulation induces a selective activity suppression within the co-tuned network for processing their preferred frequency.

Together, these results indicate that the effect of holographic optogenetic stimulation depends not on the specific tuning of cells, but on the co-tuning between stimulated and non-stimulated neurons. Also, this effect is not driven solely by a few non-target cells with large response changes. Rather, the overall population of cells shows relative response changes due to the stimulation when synchronized with their preferred frequency.

Overall, these results further suggest that when neural activity is increased in a subset of target cells due to photostimulation in addition to the target sound presentation, other co-tuned cells selectively reduce their tone-evoked responses to their preferred tone presentation, indicating that the network rebalances to maintain network activity within a certain range.

## Rebalanced network responses are stable

We then questioned whether such response amplitude changes due to stimulation within the local network are persistent. To test whether the rebalanced status of the neuronal ensemble is persistent, we examined the tone-evoked response amplitude changes between the post-stimulation and the stimulation sessions $\left(\text{post} - \text{stimulation effect}: \Delta F/F_{(post-stimulation\ session)} - \Delta F/F_{(stimulation\ session)}\right)$. Response

amplitudes were similar across conditions and tone presentation frequencies for both groups of cells ($F$(2, 4056)=1.83, p=0.16; *Figure 3E*). These results indicate that pairing exogenous stimulation on a subset of neurons along with sounds can instantaneously change the network responses to sounds, and this change can persist at least for many minutes during the experimental session.

## Neurons with higher frequency selectivity show greater response changes

Our results demonstrate that response changes on non-target cells are significantly influenced by the frequency tuning of stimulation-target cells as well as the frequency of the presented pure tone along with the stimulation. However, we also observed a marginal stimulation effect in the 54 kHz non-target cell group during 54 kHz pure tone presentation, even when 16 kHz target cells were stimulated. We reasoned that this effect might be due to some weak sound activation of 54 kHz cells by 16 kHz tones potentially due to the asymmetric shapes of many auditory tuning curves in AC (*Schreiner et al., 2000*; *Liu and Kanold, 2021*). Indeed, many cells exhibited broad tuning properties, responding to both 16 and 54 kHz (*Figure 3B*). Thus, this marginal stimulation effect could be attributed to cells grouped as 54 kHz preferring cells, yet still showing sound-evoked responses to 16 kHz, particularly given that 16 kHz is within the sensitive frequency range in mice (*Kane et al., 2012*).

Building on our findings of a rebalanced cortical network, we next aimed to identify whether frequency tuning selectivity influences response amplitude changes in the non-targeted co-tuned neurons. For each cell, we calculated the frequency preference index $\left(\Delta F/F_{(16kHz)}\right) - \left(\Delta F/F_{(54kHz)}\right)$ and divided the cells into three categories of frequency selectivity: low, mid, and high. We removed cells with extreme frequency preference values, where the index values exceed ±4 standard deviations (SDs) from the median, prior to dividing them into three categories. This removed about 1% of cells from the dataset for further analyses. This grouping was based on the 33% quartile ranges, with each category representing one-third of the data distribution (*Figure 3B*). Values closer to 0 indicate more broadly tuned cells across frequencies while extreme positive and negative values indicate sharply tuned cells to either frequency.

We then tested whether cells with higher frequency selectivity to one frequency exhibited greater response amplitude changes. We performed a three-way ANOVA to examine the effect of frequency selectivity (low, mid, high selectivity), stimulation condition (control, 16 kHz target stim, 54 kHz target stim), and cell groups (16 kHz vs. 54 kHz preferring cells) on the response amplitude change. There were significant main effects of frequency selectivity ($F$(2, 2183)=23.52, p<0.0001) and stimulation condition ($F$(2, 2183)=11.03, p<0.0001). No significant main effect of cell group was observed ($F$(1, 2183)=0.77, p=0.379). Thus, neither the interaction between frequency selectivity and cell group ($F$(2, 2183)=0.69, p=0.503), nor the interaction between condition and cell group ($F$(2, 2183)=2.64, p=0.072) was significant. The three-way interaction between frequency selectivity, stimulation condition, and cell group was also not significant ($F$(4, 2183)=0.86, p=0.487).

However, the interaction between frequency selectivity and stimulation condition was significant ($F$(4, 2183)=2.82, p=0.0238), indicating that the effect of frequency selectivity depended on the condition. These results suggest that the response amplitude changes across conditions were more prominent for cells with higher frequency selectivity.

To identify where the significant response difference occurred across conditions, we further performed post hoc pairwise comparisons between conditions within each frequency selectivity category for each cell group. For 16 kHz preferring cells, we observed a significant difference in the response change between control and 16 kHz stim conditions only from the high-frequency selectivity category (p<0.027, Holm–Bonferroni correction for multiple comparisons). In parallel, for 54 kHz preferring cells, the significant effect was observed between control and 54 kHz stim conditions (p=0.033, Holm–Bonferroni correction for multiple comparisons; *Figure 3F*). These results indicate that non-targeted cells with higher frequency selectivity exhibit the greatest response amplitude changes only when the target-stimulated cells were co-tuned with them (*Figure 3—figure supplement 2*). These results also suggest that frequency-selective neurons form co-tuned networks.

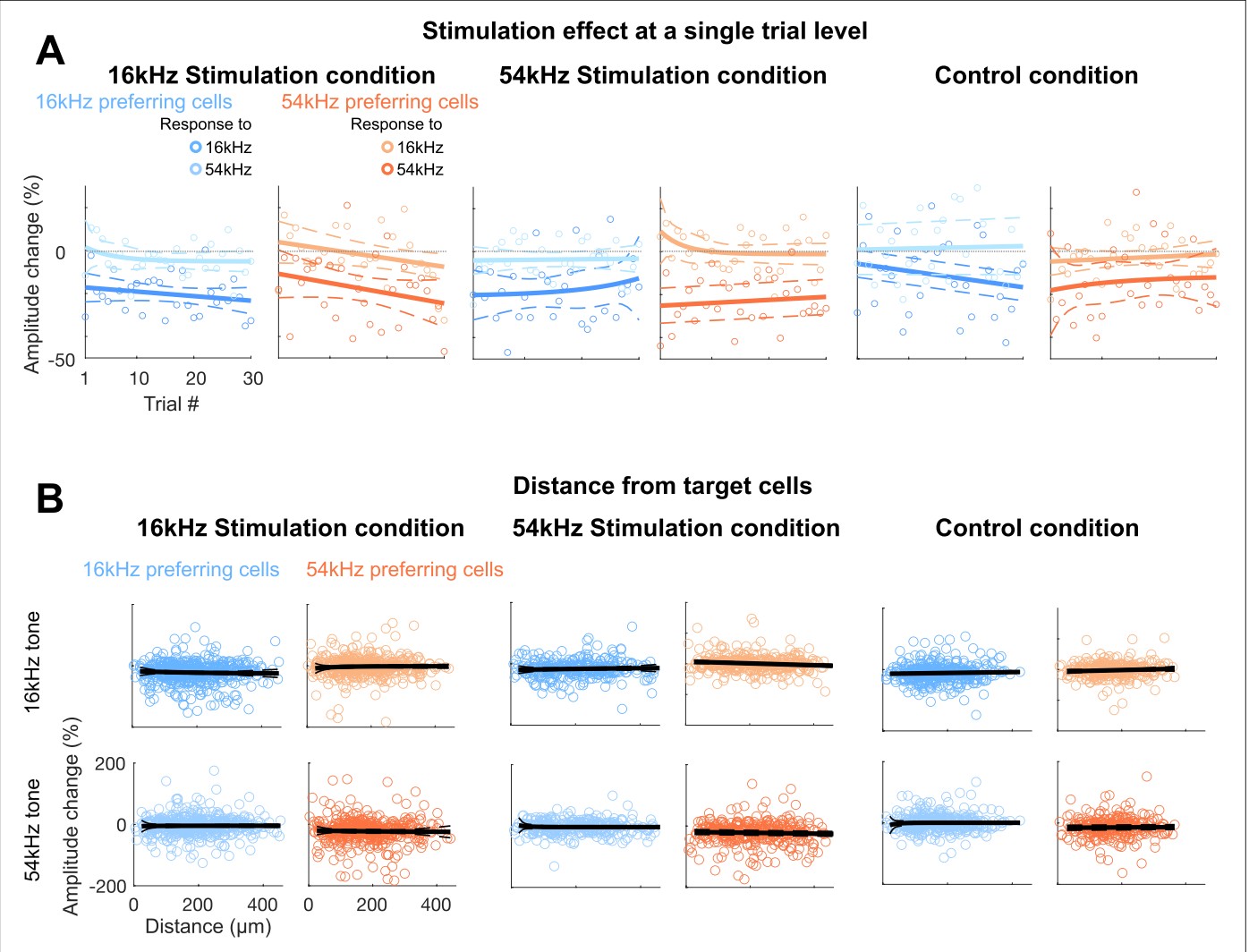

**Figure 4.** Rebalanced response changes on non-target 16 kHz cells are immediate and widely distributed. (**A**) Stimulation effect ($\Delta F/F_{stim} - \Delta F/F_{baseline}$) in 16 kHz (blue) and 54 kHz (orange) preferring cells per each trial for the 5 cell 16 kHz stimulation condition (left), 54 kHz stimulation condition (middle), and the no-cell control condition (right). Each circle represents average stimulation effect per each trial. Decreased amplitudes to preferred frequencies were observed from as early as trial 1 with no significant further changes across trials, regardless of frequencies and conditions (sum-of-squares *F*-test, all $p > 0.05$). Solid lines indicate fitted curves and dashed lines indicate 95% confidence intervals. Vertical gray dashed lines indicate 0 amplitude change. (**B**) Stimulation effect ($\Delta F/F_{stim} - \Delta F/F_{baseline}$) of each non-target 16 kHz (blue) or 54 kHz (orange) preferring cells for either 16 kHz (top row) or 54 kHz (bottom row) presentation in relation to the mass of center distance to any target cells for the stimulation condition (left), 54 kHz stimulation condition (middle), and the control condition (right). Each circle represents each cell. For the control condition, we considered the top 5 most tone-responsive cells from the baseline session as 'target' cells, as there was no stimulation. Non-target cells are widely distributed within the field of view (FOV) (550 µm²), regardless of cell groups, frequencies, and conditions. Gray lines indicate fitted curves, excluding cells that are closer than 15 µm (vertical green lines; cells <15 µm marked in lighter shades), and dashed lines indicate 95% confidence intervals.

## Sparsely distributed non-target co-tuned ensembles immediately rebalance their activities to maintain the network balance

Network balance can be achieved by multiple mechanisms operating on different timescales. To get insight into the potential mechanisms underlying the observed rebalancing, we next investigated how rapidly cells start adjusting their responses during the stimulation condition. We thus examined the stimulation effect (changes in response amplitude due to stimulation of target cells) for non-target co-tuned ensembles at the single-trial level. We observed decreased response amplitudes from the first trial, with no significant decay across trials (*Figure 4A*), regardless of cell groups, frequency presentation, and conditions (sum-of-squares *F*-test, all $p > 0.05$). The absence of trial-related response

amplitude changes in non-target co-tuned ensembles indicates that non-target co-tuned cells immediately change their activity whenever targeted cells increased their activity due to stimulation, to maintain the network balance.

## Non-target co-tuned ensembles that show rebalancing are spatially distributed

Activity rebalancing could be driven by local, for example, changes in excitatory-inhibitory (E/I) balance (*Froemke, 2015*), or distributed changes. To identify whether co-tuned ensembles that changed their activities are locally or widely distributed, we computed the center of mass distance between each non-target cell to any of the target cells. For the stimulation condition, we excluded non-target cells that were within 20 μm distance of the target cells to ensure that any effects from those neighboring cells with their increased response amplitudes could have been not, even partially, due to photostimulation (*Figure 1B–D*). We observed that non-target co-tuned ensembles were widely distributed within the FOV, similar to non-target non-co-tuned ensembles as well as those from the control condition (sum-of-squares *F*-test, p>0.05; *Figure 4B*). This indicates that activity changes of non-target co-tuned ensembles are not merely the result of direct input from external photostimulation within a tight localized network. Rather, widespread, sparsely represented co-tuned ensembles continuously update incoming information based on their tuning properties.

## Discussion

Trial-by-trial variability in neuronal activity is ubiquitous in the brain, with sensory stimuli evoking activity in different sparse co-tuned ensembles at different times. How sensory-evoked activity is distributed and coordinated across sparsely distributed co-tuned networks has been unknown. Here, we leveraged the capability for selective in vivo stimulation via holographic optogenetics to investigate how functionally related neuronal ensembles in AC coordinate activity. Our results show that manipulating a small subset of target co-tuned neurons alters the auditory-evoked responses of other non-target co-tuned neurons. Increased activity in one subset of neurons is balanced by decreased activity in the rest of the co-tuned population of neurons.

Importantly, such network rebalancing occurs only for processing acoustic features specific to their tuning. Our analysis shows that the most selective non-targeted neurons show the strongest effect after stimulation, suggesting that selective neurons form functionally interacting sub-networks consistent with in vivo correlation analyses (*Rupasinghe et al., 2021*) and in vitro studies in VC (*Ko et al., 2014*). Functionally related cells might form these subnetworks during development likely due to lineage relationships and Hebbian processes (*Katz and Shatz, 1996*; *Ohtsuki et al., 2012*; *Li et al., 2012*). Together, our findings suggest that neuronal ensembles with strengthened connectivity across neuronal ensembles sharing similar functional tuning properties actively interact and update their status to maintain the overall network, enabling energy-efficient sensory processing.

The present work applied holographic optogenetic stimulation to manipulate neuronal activity at a single-cell resolution in the AC for the first time. Similar to previous findings in VC (*Chettih and Harvey, 2019*; *LaFosse et al., 2024*), our study further supports the idea that extra activation within the network exerts an inhibitory influence on a subset of neurons. In the VC, single-cell stimulation resulted in suppression of neighboring co-tuned neurons (*Chettih and Harvey, 2019*). Our results here show that feature-specific suppression occurs in spatially dispersed ensembles of non-target co-tuned neurons. The widely distributed response amplitude decreases in those neurons suggest that this phenomenon is not limited to local neighboring cells but involves widespread networks (*Bowen et al., 2020*). Thus, the effect is not solely due to inhibition caused by neighboring interneurons from the optogenetic stimulation. Instead, neurons with similar functional characteristics, sparsely distributed throughout the AC, actively interact with more sharply tuned neurons to modulate their activity the most.

Cortical networks are shaped by dynamic changes in neural activity driven by various factors. Neurons rapidly modulate their responses based on their functional roles in sensory processing. Recurrent cortical networks are thought to update their activity based on incoming information to maintain homeostatic balance (*Karmarkar and Buonomano, 2007*; *Liu et al., 2023*). Concurrently, co-activated neurons processing similar acoustic properties strengthen their connectivity by Hebbian

learning (*Hebb, 2005*; *Malenka and Nicoll, 1999*; *Brown et al., 1990*). Thus, cortical networks are shaped by an active interplay of synaptic plasticity, homeostasis, and Hebbian learning, rather than by a single dominant mechanism (*Vitureira and Goda, 2013*; *Fox and Stryker, 2017*; *Toyoizumi et al., 2014*; *Turrigiano and Nelson, 2000*). Rebalancing of network activity is often attributed to homeostatic rebalancing of individual cell's activity (*Liu et al., 2023*; *Yin and Yuan, 2014*) or an E/I balance: increased activity of inhibitory neurons resulting in reduced activity of excitatory neurons (*Froemke, 2015*). Our results suggest that rebalancing is tuning-specific.

Given that inhibitory neurons in A1 are generally less frequency selective than excitatory neurons (*Maor et al., 2016*), changes in inhibition are unlikely to be the only contributor to the observed effect. While our AAV was not cell-type specific, it is also unlikely that many selected target neurons would be inhibitory interneurons as a greater proportion of neurons in sensory cortices are excitatory (about 80% compared to 20% inhibitory *Wang et al., 2018*; *Marin, 2012*; *Xu et al., 2016*; *Wonders and Anderson, 2006*; AAV-hSyn is expressed in both excitatory and inhibitory neurons in similar proportions *Nathanson et al., 2009*). Furthermore, our primary cell selection criterion for stimulation yielded a subgroup of strong specific frequency-responsive cells (top 30% of cells that show the biggest evoked responses to the target tone after excluding cells that show responses to multiple tones). This criterion likely selected more excitatory neurons, as they generally show greater stimulus-selective responses than inhibitory neurons in sensory cortices (*Maor et al., 2016*; *Kerlin et al., 2010*).

It is noteworthy that not all target neurons showed clear activation in response to holographic stimulation. Although we attempted to pre-select cells responsive to stimulation, some of them seemed to exhibit reduced activation during the experiment, potentially due to motion artifacts, response adaptation, network suppression, or trial-by-trial variability. Nevertheless, the frequency-specific suppression on co-tuned neurons observed in this study suggests that this effect can be driven by activation of a very small number of neurons. One caveat is that, while we presume most target cells were likely excitatory, inhibitory neurons may have been included in the target cell group. Inhibitory neurons could show a reliable, strong response to the optogenetics compared to more variable responses that excitatory neurons could show (*Mateo et al., 2011*), thus may have been included in the target cell group. However, if more inhibitory cells were included, this would have reduced trial-by-trial variability from the stimulation yielding higher probability of target cell activation, which is different from what we observed. Additionally, reoccurring sounds evoke activity changes in both excitatory and inhibitory neurons in the same direction, leaving the E/I balance unchanged (*Wehr and Zador, 2003*; *Chen et al., 2015*). Moreover, the fraction of inhibitory cells is 10–20% of all cortical neurons (*Meyer et al., 2011*; *Tasaka et al., 2023*; *Markram et al., 2004*), thus the chance of stimulating them is small. Together, these considerations suggest that our results are unlikely to be the effect of the activation of inhibitory neurons. Rather, other balancing mechanisms such as short-term depression at thalamocortical synapses may play a role (*Li and Glickfeld, 2023*).

In contrast to slow changes of homeostatic plasticity related to the E/I balance, plasticity in cortical cellular responses can occur quickly and is cell-specific, thereby tuned to their functional response properties (*Davis, 2013*; *Tien et al., 2017*; *Maffei and Fontanini, 2009*). Based on this, we speculate that the decrease in response amplitudes of non-target neurons likely reflects rapid activity-dependent synaptic changes in excitatory cells. Future work using Cre-dependent virus expression or a cell type-specific labeling approach will be required to confirm cell-type-specific roles in this phenomenon and underlying mechanisms.

Lastly, no additional response changes were observed in the post-stimulation session, indicating that the rebalanced network status remained constant. Indeed, constant amplitudes were observed regardless of conditions, subgroups, and frequencies, suggesting that the network persistence was achieved through repeated acoustic stimuli presentation and photostimulation. The persistence after the stimulation condition further suggests that once the newly learned rebalanced network status is achieved, the network response stabilizes and remains persistent. The mechanisms behind this stabilization are unclear but may involve an active interplay of homeostasis and Hebbian learning to form co-active networks (*Barak and Tsodyks, 2007*).

Taken together, the present study reveals how neuronal ensembles in the AC rebalance to maintain a homeostatic processing equilibrium for a given sensory input, and that rebalanced networks remain persistent. Moreover, our results show that network activity can be controlled by even a small subset of neurons, and the network changes are closely tied to the functional tuning properties of neurons.

## Materials and methods

### Animals

A total of 14 mice over 8 weeks old (8–30 weeks, 6 males and 8 females) were used in the experiments. To retain high-frequency hearing at experimental ages, offspring from C57BL6/J and B6.CAST-*Cdh23*[Ahl/+]/Kjn (Jax 002756) were used for all experiments. Animals were housed on a 12 h reversed light/dark cycle. All experimental procedures were approved by Johns Hopkins Institutional Animal Care and Use Committee (protocol# MO23M146).

### Preparation of AAV9-hSyn-GCaMP8s-T2A-rsChRmine

To generate the AAV9-hSyn-GCaMP8s-T2A-rsChRmine virus, a gene fragment containing T2A, rsChRmine, Kv2.1 soma-targeted localization motif, and 3xHA tag (synthesized by Twist Biosciences) was subcloned into the pGP-AAV-syn-jGCaMP8s plasmid vector (Addgene: 162374). Viral vectors were commercially prepared (Virovek) to a concentration of $1 \times 10^{13}$ vg/mL.

### Surgery and virus injection

Surgery was performed as described in previous studies (*Francis et al., 2018*; *Francis et al., 2022*). We injected dexamethasone (1 mg/kg, VetOne) subcutaneously (s.c.) to minimize brain swelling prior to surgery. 4% isoflurane (Fluriso, VetOne) with a calibrated vaporizer (Matrix VIP 3000) was used to induce anesthesia, which was then reduced down to 1.5–2% for maintenance. Body temperature was monitored and maintained at around 36°C throughout the surgery (Harvard Apparatus Homeothermic monitor). We first removed the hair on the head using a hair removal product (Nair) to expose the skin. Betadine and 70% ethanol were applied three times to the exposed skin. Skin and tissues were then removed, and muscles were scraped to expose the left temporal side where the craniotomy was conducted. Unilateral craniotomy was performed to expose about 3.5 mm diameter region over the left AC. Virus (AAV9-hSyn-GCaMP8s-TSA-rsChRmine titer of 1:2) injection was performed at 2–3 sites near tentative A1 area at about 300 μm depth from the surface, using a glass pipette controlled by a micromanipulator (Sutter Instrument MPC-200 and ROE-200 controller). We injected 300 nL on each site at the rate of 180 nL/min (Nanoject3). Once virus injection was completed, two circular glass coverslips (one of 3 mm and one of 4 mm in diameter) were affixed with a clear silicone elastomer (Kwik-Sil, World Precision Instruments). An extra layer of dental acrylic (C&B Metabond) was applied around the edge of the cranial window to further secure it, cover the exposed skull, and adhere a custom 3D-printed stainless steel headpost. Carprofen (5 mg/kg) and cefazolin (300 mg/kg) were injected (s.c.) postoperatively. Animals were given at least 3–4 weeks of recovery and viral expression time before any imaging was performed.

### Experimental procedures

Fully awake animals were head-fixed on a custom-made stage, where a free field speaker (TDT ED1) was faced toward the right ear at 45°. All sound stimuli were driven by TDT RX6 multiprocessor and we imaged GCaMP8s responses with a resonant scanning two-photon microscope (Bruker Ultima 2Pplus; 940 nm excitation wavelength) at A1 L2/3 (160–200 μm). A1 was identified by its tonotopy gradient using widefield imaging (*Figure 1E*; *Liu et al., 2019*). During two-photon imaging sessions, we first conducted a short imaging session (about 1 min) presenting 100 ms pure tones at three different frequencies (4, 16, 54 kHz) at 70 dB sound pressure level (SPL), covering the mouse hearing range, for 10 times each in a randomized order (ISI: 2 s). This was to identify initial tuning properties of sound-responsive cells (cell selection session; *Figure 1E*). The acquired imaging data was immediately processed using 'suite2p' and if the mean trace during tone presentation exceeded 2 SDs of the mean trace during the 300 ms period before sound onset as a baseline. We continued only when at least 50% of cells within the FOV showed sound-evoked responses. We took 16 or 54 kHz as our target functional properties. Target frequency was randomly assigned. We manually tested the response changes to stimulation of the top 30% of target frequency responsive cells (~20 cells) selected from the cell selection session by using a stimulation laser (Light Conversion Carbide; 1040 nm excitation wavelength). Stimulation laser power was set around 5 mW per cell. We selected five representative cells (target cells) that showed visible fluorescence changes to the stimulation.

Prior to the main experimental session, we ran a short (~1 min) stimulation session without any sound presentation that comprised five trials of 100 ms stimulation with 6 s ISIs for a rapid check of

the stimulation effect. This session was restricted to five trials of stimulation given the limited time of the imaging session, leading to larger variability of the observed stimulation effect. We further verified the stimulation effect from the experimental stimulation session where 30 trials were given.

For the main experiment, three consecutive imaging sessions were followed by the presentation of either 16 or 54 kHz 100 ms pure tones with random ISIs between 5.8 and 6.5 s for 30 trials each, as baseline session, stimulation session, and post-stimulation session (*Figure 1E*). Only the second imaging session (i.e., stimulation session) received holographic stimulation on the pre-selected five target cells for 30 revolutions of 15 µm spiral for about 100ms at 5 mW laser power per cell (16 kHz target cell or 54 kHz target cell stimulation conditions) or 0 mW laser power per cell (control condition). The default mechanical setup (Bruker PrairieView version 5.7) of the microscope opens and closes the uncaging shutter to enable the stimulation laser path for every single stimulation time point, which causes an external mechanic sound that can trigger neural activation of the AC. To minimize any effect of external mechanic sounds to cells in the AC, we kept the uncaging shutter open during the imaging session. The number of imaging sessions per animal varied depending on the virus expression. Regardless, all animals were used for control and at least one stimulation condition with a minimum of 2 days apart by varying FOVs to avoid imaging the same cells multiple times. The order of conditions and the imaging depth presented to the same animal were randomized.

An additional single-cell stimulation only session was conducted on a subset of animals to further test a reliable holographic stimulation at a single-cell level (n=3 animals). We varied a stimulation position from the original target stimulation point to 10, 20, or 30 µm shifted along the x-axis or y-axis, generating seven different stimulation points (original cell position, 10, 20, or 30 µm shifted along the x-axis, 10, 20, or 30 µm shifted along the y-axis). We stimulated each stimulation point for 10 times in a randomized order across stimulation points with 8 s ISIs.

## Analysis

Imaging data were processed with 'suite2p' for motion correction, cell detection, and cell fluorescence trace extraction (*Pachitariu et al., 2016*). Any further analyses were done using Matlab (R2020b) (*Kang et al., 2025*). We applied neuropil correction to the fluorescence traces using the following equation: $F_{(cell\_corrected)} = F_{(cell)} - (0.8 * F_{(neuropil)})$. We then computed ΔF/F normalized to the baseline, by following the equation: $\Delta F/F = (F_{(trace)} - F_{(baseline)})/F_{(baseline)}$, where baseline is about 300ms before the sound onset. For the single-cell level stimulation session, we computed peak ΔF/F per each stimulation point and applied a mixed-effect model by taking peak ΔF/F as dependent variables, stimulation point as independent variables, and cells as random factor. For the five-cell stimulation validation session, we computed a proportion of activated cells (any cell with peak ΔF/*F*>0) among stimulated cells per each FOV as well as peak ΔF/F and ran a random permutation with 100 iterations.

For experimental sessions, to select sound-responsive cells, we compared sound-evoked activity (160 ms – 660 ms after sound onset capturing the sound-evoked response due to slow calcium transient) and the baseline activity (300 ms – 0 ms before the sound onset). We considered cells sound responsive when the amplitude of the average sound-evoked activity exceeded two standard deviations of the amplitude of the average baseline activity. Sound-responsive cells were selected based on fluorescence traces only from the baseline session to minimize any potential effect of stimulation on cell selection. We then computed the ratio of the evoked activity between two frequencies as an index of the frequency preference $\left(\Delta F/F_{(16kHz)} - \Delta F/F_{(54kHz)}\right)$. We subgrouped sound-responsive cells into either 16 kHz preferring cells or 54 kHz preferring cells based on the frequency preference, taking 0 as a subgroup criterion. As our main interest was changes to non-target cells, we excluded target cells for further analyses. To compare response changes due to stimulation for each group of cells, average sound-evoked activity of sound-responsive cells from the baseline session was subtracted from the stimulation session $\left(\text{stimulation effect}: \Delta F/F_{(stimulation\ session)} - \Delta F/F_{(baseline\ session)}\right)$ for each condition. We further compared response changes between post-stimulation session and stimulation session, again by subtracting the response amplitudes between two sessions for each group and condition $\left(\text{post} - \text{stimulation effect}: \Delta F/F_{(post-stimulation\ session)} - \Delta F/F_{(stimulation\ session)}\right)$. To quantify the stimulation effect based on functional properties for conditions and groups, we applied a mixed-effect model by taking amplitude changes as dependent variables, frequency, conditions, and cell groups as independent variables, and FOVs as random factor. We then computed type III ANOVA to examine whether the effect of functional property (frequency) to the amplitude changes was specific

to the target frequency presentation synchronized with the stimulation (i.e., 16 for 16 kHz target cell stimulation or 54 for 54 kHz target cell stimulation). To quantify whether the response amplitude changes due to stimulation differ across trials, we fitted a dataset of average stimulation effect across each trial per each condition (stimulation or control), each cell group (16 kHz preferring or 54 kHz preferring cells), and each tone presentation (16 or 54 kHz) to the three-parameter model and computed the extra sum-of-squares $F$-test to compare whether response amplitude changes across trials were different from a constant line (**Motulsky and Christopoulos, 2004**). To quantify a relationship between response amplitude changes of non-target cells and their distances to target cells, we computed a center of mass distance of each cell position relative to target cells and fitted the dataset of the response amplitude changes across distance to the three-parameter model to compute the extra sum-of-squares $F$-test, per each condition (16 kHz target stimulation, 54 kHz target stimulation, or control), each cell group (16 kHz preferring or 54 kHz preferring cells), and each tone presentation (16 or 54 kHz). For the control condition, as there were no stimulated target cells, we chose the top five most tone-responsive cells from the baseline session as 'target' cells.

We then generated a simple model in which a suppression term was applied either to all neurons or specifically to non-target co-tuned cells to test our results from the data. We took a similar range of number of neurons and FOVs to closely simulate the model to the real dataset structure. On 50 neurons (n) per FOV across 18 FOVs, the simulated calcium trace of each neuron was defined as

$$\mathrm{Trace_{n(t)} = R_{n(t)} - theta_n + epsilon_{n(t)},}$$

where $R_{n(t)}$ is a time-varying response amplitude from the sound onset to the offset, modeled separately for the baseline and stimulation sessions. The suppression term $theta_n$ was applied only during the stimulation session either to all neurons, randomly selected neurons, or only non-target co-tuned neurons depending on the simulation condition, and $epsilon_{n(t)}$ is additive Gaussian noise. To simulate sound-evoked calcium transients, we assigned a faster decay time constant (200ms) for non-target co-tuned neurons $R_{n(t)}$ and a slower decay (1000ms) for non-target non-co-tuned neurons $R_{n(t)}$ for both the baseline and stimulation sessions. Theta was defined as proportional to the average stimulation strength from target neurons, derived from the real dataset, and scaled by a factor $\alpha=0.3$ in the current simulation. To introduce neuron-level variability, an additional jitter ($epsilon_n$) was applied as follows:

$$\mathrm{Theta_n = \alpha^* \ mean(target \ stimulation \ amplitude)^*(1 + epsilon_n).}$$

Similar to the real data analyses, we compared the response change between the stimulation and baseline sessions' trace amplitudes.

## Histology

Animals were deeply anesthetized with 4% isoflurane to perform transcardial perfusion with 4% paraformaldehyde (PFA) in 0.1 M phosphate buffer saline (PBS). The extracted brains were post-fixed in 4% PFA for an additional 12–24 h. Coronal brain sections at 50 μm containing the AC were stained with primary antibodies of HA-Tag (1:500) and chicken Green Fluorescent Protein (GFP, 1:500) for GCaMP8s, and secondary antibodies of 594-conjugated anti-rabbit IgG (1:1000) and 488-conjugated anti-chicken IgG (1:1000) for red-shifted opsins.

## Acknowledgements

This study was supported by U19 NS107464 (POK), NIH RO1DC017785 (POK), and NIH F32DC019842 (TAB).

## Additional information

### Funding

| Funder | Grant reference number | Author |
|---|---|---|
| National Institute of Neurological Disorders and Stroke | U19NS107464 | Patrick O Kanold |
| National Institute on Deafness and Other Communication Disorders | R01DC017785 | Patrick O Kanold |
| National Institute on Deafness and Other Communication Disorders | F32DC019842 | Travis A Babola |

The funders had no role in study design, data collection and interpretation, or the decision to submit the work for publication.

### Author contributions

HiJee Kang, Conceptualization, Data curation, Software, Formal analysis, Validation, Investigation, Visualization, Methodology, Writing – original draft, Writing – review and editing; Travis A Babola, Conceptualization, Data curation, Validation, Investigation, Visualization, Methodology, Writing – original draft, Writing – review and editing; Patrick O Kanold, Conceptualization, Resources, Supervision, Funding acquisition, Methodology, Writing – original draft, Project administration, Writing – review and editing

### Author ORCIDs

HiJee Kang ● https://orcid.org/0000-0002-9037-975X
Travis A Babola ● https://orcid.org/0000-0003-4440-5029
Patrick O Kanold ● https://orcid.org/0000-0002-7529-5435

### Ethics

All experimental procedures were approved by Johns Hopkins Institutional Animal Care and Use Committee (protocol #MO23M146).

Reviewer #1 (Public review): https://doi.org/10.7554/eLife.104242.4.sa1
Reviewer #2 (Public review): https://doi.org/10.7554/eLife.104242.4.sa2
Author response https://doi.org/10.7554/eLife.104242.4.sa3

## Additional files

### Supplementary files

MDAR checklist

### Data availability

All preprocessed imaging data and relevant analyses scripts are deposited at Johns Hopkins University Research Data Repository.

The following dataset was generated:

| Author(s) | Year | Dataset title | Dataset URL | Database and Identifier |
|---|---|---|---|---|
| Kang H, Babola TA, Kanold PO | 2025 | Data and code associated with the publication: Rapid rebalancing of co-tuned ensemble activity in the auditory cortex | https://doi.org/10.7281/T1RZQICN | Johns Hopkins Research Data Repository, 10.7281/T1RZQICN |

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
