## [Editor Report · eLife Assessment]

This study uses all-optical electrophysiology methods to provide a **valuable** insight into the organization of cortical networks and their ability to balance the activity of groups of neurons with similar functional tuning. The all-optical approach used in this study is impressive, and the claim that the effects of optical stimulation correspond to a specific homeostatic mechanism is **solid**. The work will be of interest to neurobiologists and to developers of optical approaches for interrogating brain function.

---

## [Referee Report · Reviewer #1 (Public review)]

Summary:

Kang et al. provide the first experimental insights from holographic stimulation of auditory cortex. Using stimulation of functionally-defined ensembles, they test whether overactivation of a specific subpopulation biases simultaneous and subsequent sensory-evoked network activations.

Strengths:

The investigators use a novel technique to investigate the sensory response properties in functionally defined cell assemblies in auditory cortex. These data provide the first evidence of how acutely perturbing specific frequency-tuned neurons impacts the tuning across a broader population. Their revised manuscript appropriately tempers any claims about specific plasticity mechanisms involved.

Weaknesses:

Although the single cell analyses in this manuscript are comprehensive, questions about how holographic stimulation impacts population coding are left to future manuscripts, or perhaps re-analyses of this unique dataset.

---

## [Referee Report · Reviewer #2 (Public review)]

The goal of HiJee Kang et al. in this study is to explore the interaction between assemblies of neurons with similar pure-tone selectivity in mouse auditory cortex. Using holographic optogenetic stimulation in a small subset of target cells selective for a given pure tone (PTsel), while optically monitoring calcium activity in surrounding non-target cells, they discovered a subtle rebalancing process: co-tuned neurons that are not optogenetically stimulated tend to reduce their activity. The cortical network reacts as if an increased response to PTsel in some tuned assemblies is immediately offset by a reduction in activity in the rest of the PTsel-tuned assemblies, leaving the overall response to PTsel unchanged. The authors show that this rebalancing process affects only the responses of neurons to PTsel, not to other pure tones. They also show that assemblies of neurons that are not selective for PTsel don't participate in the rebalancing process. They conclude that assemblies of neurons with similar pure-tone selectivity must interact in some way to organize this rebalancing process, and they suggest that mechanisms based on homeostatic signaling may play a role.

The authors have successfully controlled for potential artefacts resulting from their optogenetic stimulation. This study is therefore pioneering in the field of the auditory cortex (AC), as it is the first to use single-cell optogenetic stimulation to explore the functional organization of AC circuits in vivo. The conclusions of this paper are very interesting. They raise new questions about the mechanisms that could underlie such a rebalancing process.

(1) This study uses an all-optical approach to excite a restricted group of neurons chosen for their functional characteristics (their frequency tuning), and simultaneously record from the entire network observable in the FOV. As stated by the authors, this approach is applied for the first time to the auditory cortex, which is a tour de force. However, such approach is complex and requires precise controls to be convincing. The authors provide important controls to demonstrate the precise ability of their optogenetic methods. In particular, holographic patterns used to excite 5 cells simultaneously may be associated with out-of-focus laser hot spots. Cells located outside of the FOV could be activated, therefore engaging other cells than the targeted ones in the stimulation. This would be problematic in this study as their tuning may be unrelated to the tuning of the targeted cells. To control for such effect, the authors have decoupled the imaging and the excitation planes, and checked for the absence of out-of-focus unwanted excitation (Suppl Fig1).

(2) In the auditory cortex, assemblies of cells with similar pure-tone selectivity are linked together not only by their ability to respond to the same sound, but also by other factors. This study clearly shows that such assemblies are structured in a way that maintains a stable global response through a rebalancing process. If a group of cells within an assembly increases its response, the rest of the assembly must be inhibited to maintain the total response.

The boundary between assemblies is smooth as the rebalancing process occurring in one assembly seem to affect also the response of the other assembly comprising cells tuned to a the other frequency. This trend is not significant but visible for both tested frenquencies in Fig. 3 and Fig S3.

---

## [Author Response]

The following is the authors’ response to the previous reviews

**Reviewer #1 (Public review):**
Summary:Kang et al. provide the first experimental insights from holographic stimulation of auditory cortex. Using stimulation of functionally-defined ensembles, they test whether overactivation of a specific subpopulation biases simultaneous and subsequent sensory-evoked network activations.Strengths:The investigators use a novel technique to investigate the sensory response properties in functionally defined cell assemblies in auditory cortex. These data provide the first evidence of how acutely perturbing specific frequency-tuned neurons impacts the tuning across a broader population. Their revised manuscript appropriately tempers any claims about specific plasticity mechanisms involved.Weaknesses:Although the single cell analyses in this manuscript are comprehensive, questions about how holographic stimulation impacts population coding are left to future manuscripts, or perhaps re-analyses of this unique dataset.
**Reviewer #2 (Public review):**
The goal of HiJee Kang et al. in this study is to explore the interaction between assemblies of neurons with similar pure-tone selectivity in mouse auditory cortex. Using holographic optogenetic stimulation in a small subset of target cells selective for a given pure tone (PTsel), while optically monitoring calcium activity in surrounding non-target cells, they discovered a subtle rebalancing process: co-tuned neurons that are not optogenetically stimulated tend to reduce their activity. The cortical network reacts as if an increased response to PTsel in some tuned assemblies is immediately offset by a reduction in activity in the rest of the PTseltuned assemblies, leaving the overall response to PTsel unchanged. The authors show that this rebalancing process affects only the responses of neurons to PTsel, not to other pure tones. They also show that assemblies of neurons that are not selective for PTsel don't participate in the rebalancing process. They conclude that assemblies of neurons with similar pure-tone selectivity must interact in some way to organize this rebalancing process, and they suggest that mechanisms based on homeostatic signaling may play a role.The authors have successfully controlled for potential artefacts resulting from their optogenetic stimulation. This study is therefore pioneering in the field of the auditory cortex (AC), as it is the first to use single-cell optogenetic stimulation to explore the functional organization of AC circuits in vivo. The conclusions of this paper are very interesting. They raise new questions about the mechanisms that could underlie such a rebalancing process.(1) This study uses an all-optical approach to excite a restricted group of neurons chosen for their functional characteristics (their frequency tuning), and simultaneously record from the entire network observable in the FOV. As stated by the authors, this approach is applied for the first time to the auditory cortex, which is a tour de force. However, such approach is complex and requires precise controls to be convincing. The authors provide important controls to demonstrate the precise ability of their optogenetic methods. In particular, holographic patterns used to excite 5 cells simultaneously may be associated with out-of-focus laser hot spots. Cells located outside of the FOV could be activated, therefore engaging other cells than the targeted ones in the stimulation. This would be problematic in this study as their tuning may be unrelated to the tuning of the targeted cells. To control for such effect, the authors have decoupled the imaging and the excitation planes, and checked for the absence of out-of-focus unwanted excitation (Suppl Fig1).(2) In the auditory cortex, assemblies of cells with similar pure-tone selectivity are linked together not only by their ability to respond to the same sound, but also by other factors. This study clearly shows that such assemblies are structured in a way that maintains a stable global response through a rebalancing process. If a group of cells within an assembly increases its response, the rest of the assembly must be inhibited to maintain the total response.One surprising result is the clear boundary between assemblies: a rebalancing process occurring in one assembly does not affect the response in another assembly comprising cells tuned to a different frequency. However, this is slightly challenged by the data shown in Figure 3.Figure 3B-left, for example, shows that, compared to controls, non-target 16 kHzpreferring neurons only decrease their response to a 16 kHz pure tone when the cells targeted by the opto stimulation also prefer 16 kHz, but not when the targeted cells prefer 54 kHz. However, the inverse is not entirely true. Again compared to controls, Figure 3B (right) shows that non-target 54 kHz-preferring neurons decrease their response to a 54 kHz pure tone when the targeted cells also prefer 54 kHz; however, they also tend to be inhibited when the targeted cells prefer 16 kHz.The authors suggest this may be due to the partial activation of 54 kHz-preferring cells by 16 kHz tones and propose examining the response of highly selective neurons. The results are shown in Figure 3F. It would have been more logical to show the same results as in Figure 3B, but with the left part restricted to highly 16 kHz-selective cells and the right part to highly 54 kHz-selective cells. However, the authors chose to pool all responses to 16 kHz and 54 kHz tones in every triplet of conditions (control, opto stimulation on 16 kHz-preferring cells and opto stimulation on 54 kHz-preferring cells), which blurs the result of the analysis.

We thank reviewers for highlighting the strengths of our work and providing valuable feedback. We further developed our manuscript mainly from Reviewer 2’s point on the overall effect explained as the main result. One of the main reasons why we chose to pool all tone preferring cells instead of highly selective cells was to ensure that the observed effect not necessarily driven by only a small group of neurons but rather that the effect was driven at the population level, especially at a subject level for Figure 3B. While Figure 3F represents how highly selective cells to each frequency play a major role in the effect, we now have added additional results with only highly selective neurons as Supplementary Figure 3. The left panel shows restricting the population to highly selective neurons to 16 kHz and the right panel restricting the population to highly selective neurons to 54 kHz at cell population level to emphasize the result (Supplementary Figure 3).

We appreciate an additional raised point by Reviewer 1 regarding the stimulation effect on population coding. Our primary focus in this manuscript was to establish single cell level effects of holographic stimulation, and we believe that population coding analyses would benefit from a more cell-type-specific approach. We plan to pursue such analyses in follow-up studies where cell types can be better identified and linked to network dynamics.

**Reviewer #1 (Recommendations for the authors):**
The authors have appropriately addressed my concerns.As this dataset will be of general interest, it would be helpful to include a doi/link to their data repository in the data availability section.

Updating the data repository to the institution server is currently in progress. We will provide the correct doi or link as soon as it becomes available. In the meantime, we will ensure to share them with anyone who contacts to us directly.

**Reviewer #2 (Recommendations for the authors):**
Many references to Figures have not been updated between the two versions of the manuscript. See lines 107, 128, 297, 321 and 346.

We are sorry for the confusion with mislabelled figures. We now have updated all the figure numbers accordingly.

In the paragraph beginning on line 266, there is no explicit reference to Figure 3C.

We now added Figure 3C reference in the main text (line 290).

If the new analysis includes 15 FOV for stim on 54 kHz-preferring cells, as indicated in the rebuttal, the corresponding numbers should be corrected in lines 152 and 180.

We now updated the number of FOVs accordingly.

The added model is not explained well enough. How are the calcium traces simulated? It is difficult to ascertain whether the result shown in Figure 3C is merely a trivial consequence of the hypothesis that suppression is applied to co-tuned neurons or to all neurons.

We are sorry for the lack of important details in the explanation of the model. We simulated time-varying sound-evoked calcium transient especially by applying different decay time constant (faster decay for co-tuned neurons and slower decay for non co-tuned neurons) to closely match the real data. More detailed explanation on this is now included in the manuscript (lines 644 – 650). Since our data do not currently allow us to identify specific cell types, we focused on modelling the stronger suppression observed in co-tuned neurons, especially by adapting the stimulation effect of target cells from the real data. In this revision, we now added data showing that ‘Randomly selected cells’ from the two groups (co-tuned or non co-tuned cell groups) did not exhibit any stimulation effect (added column in Figure 3D) to further indicate that suppression specific to co-tuned neurons is the key factor underlying the observed effects in the real data. We hope to build on this work in future studies to identify cell-type-specific effects and their computational roles.

Although the rebuttal clearly states that experiments are carried out on awake animals, this information is still missing from the manuscript.

We now stated ‘Fully awake animals’ in the experimental procedures.